drought stress; root anatomy; root system architecture; salt stress.

**Corresponding author:**
Magdalena Julkowska;
Email: mmj55@cornell.edu

**Associate Editor:**
Dr. Priya Ramakrishna

# Beyond efficiency: The multi-scale architecture of robust water transport in plants

Magdalena Julkowska[1] [iD], Guillaume Lobet[2] and Arjun Chandrasekhar[3]

[1]Boyce Thompson Institute for Plant Research, USA; [2]UCLouvain, Belgium; [3]Department of Math and Computer Science, Southwestern University, USA

### Abstract

Agronomic research has long prioritized efficiency – optimizing resource use to maximize yield under stable conditions. However, as climate variability intensifies, efficiency alone might be insufficient to sustain agricultural production in the future. Instead, robustness – the ability to maintain function across diverse and unpredictable environments – emerges as a critical trait. Robustness is not a simple metric but an emergent property, arising from the interplay of redundancy, heterogeneity and plasticity across biological scales. We examine how the components of robustness (redundance, heterogeneity and plasticity) express themselves at the anatomical, architectural and genomic scale. A major challenge is the lack of a unified framework to measure robustness. We propose integrating empirical metrics – such as vessel grouping indices, root trait heterogeneity and gene expression plasticity – with computational models to quantify redundancy, heterogeneity and plasticity. By synthesizing insights from physiology, genomics and modelling, we outline a path towards designing crops that thrive in ideal settings and under environmental uncertainty.

## 1. Introduction

For decades, agronomic success has been tightly linked to the pursuit of efficiency – how effectively plants convert limited resources into biomass or yield. The concept of water use efficiency was initially described over 100 years ago by Briggs and Schatz (1913) and further refined by Passioura (1982), framing crop productivity as a product of three interacting components: water uptake, water transport and transpiration efficiency. Similar metrics – such as nitrogen use efficiency (Moll et al., 1982) and radiation use efficiency (Monteith & Moss, 1977) – have guided breeding and modelling efforts by focusing on input–output ratios. While optimizing for efficiency can lead to highly productive plants under specific conditions, it can also create 'one-trick ponies' – genotypes that perform well in heavily controlled environments but fail dramatically when conditions shift. This generates the yield gap, since the plants are not reaching their maximum potential in real-life scenarios.

In a fluctuating environment and changing climate, the focus on efficiency alone does not suffice to sustain plant growth (Rellán-Álvarez et al., 2016). This demands a shift in focus from efficiency alone to robustness – the ability to maintain function across diverse and unpredictable environments. Unlike efficiency, robustness is not easily reduced to a single number (Liu et al., 2017). It is an emergent property of biological systems, arising from interactions among redundancy, heterogeneity and plasticity across biological scales.

Water transport – spanning the soil–plant–atmosphere continuum – is an ideal system through which to explore this balance. It is inherently dynamic, subject to spatial and temporal variation in soil moisture, atmospheric demand and anatomy (Martínez-Vilalta et al., 2014). In this review, we examine how robustness in water transport manifests at the cellular and tissue level, through the root system architecture, to the genomic scales. We also take a stab at addressing a major gap in the field – how to measure robustness. While plant biologists have long studied robustness in various forms (Alseekh et al., 2025; Lachowiec et al., 2016; Masel & Siegal, 2009; Møller & Shykoff, 1999), there has been no unified framework to quantify it. Here, we propose such a framework for root systems, grounded in empirical and modelling approaches, to capture the dimensions of robustness that are critical for building resilient crops (Figure 1).

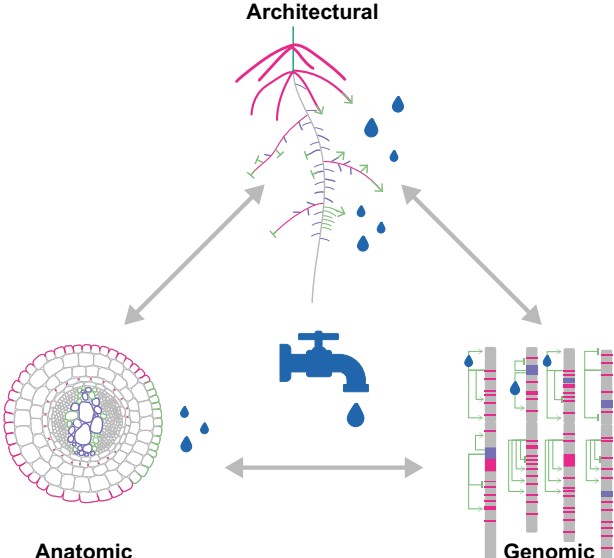

**Figure 1.** Graphical overview of principles underlying root hydraulic robustness. *Note*: The aspects related to redundancy are highlighted in pink, aspects related to heterogeneity as highlighted in purple, while aspects related to plasticity are highlighted in light green. On an anatomical level, redundancy arises from multiple conduits and barriers that maintain water flow under failure; heterogeneity reflects variation in xylem structure and tissue composition that diversifies hydraulic safety; and plasticity captures dynamic modifications in suberization and xylem differentiation that adjust conductivity to stress. On root architecture scale, redundancy is expressed through overlapping root classes and compensatory branching patterns that sustain water uptake; heterogeneity emerges from spatial and temporal diversity in root types, depths and growth angles; and plasticity encompasses environmentally driven reconfiguration of root distribution through tropisms (e.g., hydro-patterning) and growth arrest–reactivation cycles. On a genomic scale, redundancy stems from gene and genome duplications providing functional backup; heterogeneity reflects allelic and structural variation shaping population-level adaptability; and plasticity manifests as stress-responsive transcriptional and epigenetic reprogramming that enables rapid adjustment of hydraulic traits. The grey arrows indicate interactions between individual scales of root hydraulic organization.

## 2. Components of robustness

In computer science and engineering, efficiency refers to the use of minimal resources for maximal output, often resulting in streamlined designs with minimal redundancy. In contrast, robustness denotes a system's ability to maintain functionality despite perturbations or failures, often achieved through structural and functional redundancies, heterogeneous components and dynamic plasticity in response to changing conditions (Doyle et al., 2005; Kitano, 2004; Psaier & Dustdar, 2011), including physical component failures (Nelson, 1990), adversarial attacks (Muhammad & Bae, 2022) and transmission errors (Berlekamp, 1980). These design principles – redundancy, heterogeneity and plasticity – are not only hallmarks of engineered fault-tolerant systems but also critical features of plant water transport systems.

Plant hydraulic architecture must withstand a variety of stressors, including drought, salinity and mechanical damage. Understanding how redundancy, heterogeneity and plasticity act at various biological scales, from cellular traits to root system architecture and genomic organization, allows for a holistic view of robustness in plant water transport.

While we treat redundancy, heterogeneity and plasticity as distinct contributors to robustness, they are not mutually exclusive. *Heterogeneity* captures inherent or developmentally programmed variability among components – whether cells, tissues or genotypes – whereas *plasticity* describes context-dependent changes in those components triggered by environmental or physiological cues. In practice, the two can overlap – for example, stress-induced variation in vessel diameter or gene expression may appear as heterogeneity when observed across individuals, yet originates from plastic responses within each. Acknowledging this continuum clarifies that robustness arises not only from static diversity, but also from the capacity of that diversity to be reshaped through plasticity.

## 3. Laying the foundation of hydraulic functionality at the anatomical scale

At the anatomical level, the efficiency and robustness of plant water transport networks are governed by a finely tuned interplay between xylem architecture, vessel connectivity and apoplastic modifications. These structures form the physical basis for the ascent of water through plants and are shaped by evolutionary pressures to balance the demands of hydraulic conductivity with the need to withstand environmental perturbations (Bouda et al., 2022).

**Redundancy** at the tissue level is evident in the **spatial organization and grouping of xylem vessels**. Many species, especially those adapted to drought-prone environments, exhibit high vessel redundancy through the presence of multiple conduits per vascular bundle or extensive vessel interconnection (Boughalleb et al., 2014, 2015; Salih et al., 1999). This redundancy provides resilience by maintaining water flow even when individual vessels become embolized due to drought or freeze–thaw events (Carlquist, 2009). The *vessel grouping index*, a metric used to quantify the connectivity of secondary xylem vessels within a given cross-section (and that is computed as the total number of vessels divided by the total number of vessel groups – a solitary vessel counts as one vessel group), has been proposed as a proxy for hydraulic redundancy (Lens et al., 2011). High grouping indices correlate with increased resistance to total hydraulic failure, as embolism can be confined to isolated clusters rather than spreading systemically (Sperry et al., 2008).

Importantly, **different anatomical-scale processes** are redundant in themselves, providing multiple ways for the plant to reach the same functional features. For instance, decreasing root radial conductivity could be achieved by creating apoplastic barriers, but also by decreasing the production and activity of aquaporins, or a mix of both (Heymans et al., 2020). This leaves plants with multiple strategies, each with specific costs and benefits. Understanding how this diversity of mechanisms is coordinated, and under which conditions, remains an open question.

**Xylem heterogeneity** enhances robustness by diversifying the mechanical and hydraulic properties within and among tissues. Vessel diameter, wall thickness, pit membrane structure and lignin composition can vary both ontogenetically (along the root or stem axis) and across species or genotypes (Burton et al., 2013; Heymans et al., 2021; McLaughlin et al., 2024). Wider vessels generally confer greater efficiency but are more prone to embolism, while narrower vessels offer safety at the cost of reduced conductance – highlighting a classic safety-efficiency trade-off (Affortit et al., 2024; Hacke et al., 2006). Additionally, heterogeneity can emerge within a single organ. For example, metaxylem and protoxylem differ in diameter, lignification and developmental timing, providing layered responses to environmental stress. Anatomical compartmentalization into hydraulically isolated sectors – e.g., sectors with

differing vulnerability to embolism – enables partial functionality even under stress (Choat et al., 2007). This intra-plant variability is increasingly recognized as a form of structural bet-hedging, especially relevant under unpredictable water availability.

Plants also exhibit a high degree of **plasticity** in response to environmental cues. A prime example is the **modulation of apoplastic barriers**, such as suberin and lignin, in the endodermis and exodermis (Cantó-Pastor et al., 2024). These barriers regulate radial water and ion transport into the stele and are critical under drought and salinity stress. Suberization patterns can shift dynamically in response to soil water content, often increasing under drought to prevent water loss or under salt stress to restrict ion entry (Barberon et al., 2016). Plastic responses are also evident in **xylem development**. For instance, drought can accelerate xylem maturation and shift the balance between vessel size and number, reducing vulnerability to cavitation (Jang & Choi, 2018; Ramachandran et al., 2020). This developmental flexibility can be transient (reversible anatomical shifts) or involve longer-term acclimation via hormone-mediated signalling pathways, such as abscisic acid or auxin (Nieminen et al., 2015). Some of the observed variation in vessel size and structure likely reflects plastic developmental responses to environmental conditions, blurring the line between anatomical heterogeneity and plasticity.

## 4. Building below-ground robustness through root system architecture

Root system architecture defines the spatial and temporal organization of a plant's root network, directly influencing water acquisition efficiency and its robustness under environmental stress. The dynamic and hierarchical nature of root architecture – including primary, lateral or shoot-born roots – offers a rich framework for evaluating how plants achieve resilience via **redundancy**, **heterogeneity** and **plasticity** at the whole-organ level. These features shape not only how roots explore soil domains but also how they buffer against localized water scarcity or mechanical obstacles.

Root system architecture **redundancy** refers to the presence of overlapping or compensatory root structures that serve similar hydraulic functions. In monocots such as maize, the **presence of multiple shoot-born root whorls** provides hydraulic redundancy across both depth and lateral spread (Hochholdinger et al., 2004). In dicots, **extensive lateral root branching, over several topological orders,** ensures that if a portion of the primary root system is damaged or underperforms, other root segments can compensate by accessing adjacent or deeper water pools (Lynch, 2011; Schneider & Lynch, 2020). This distributed design confers robustness by minimizing the risk that water uptake is catastrophically disrupted by localized root failure or patchy drought conditions. Recently, a meta-analysis of root system hydraulic properties across more than 200 plant species has shown that the root system conductivity tends to stabilize after 60 days, despite a continuous increase in root number and growth (Baca Cabrera et al., 2024). This illustrates that, at a certain developmental point, any additional roots do not contribute to an increase in function anymore, but rather a stabilization of the existing ones.

Similar to the anatomy, the **redundancy of root developmental processes** (in addition to redundancy in structures) also ensures that identical functions can be reached in many different ways. Increasing the global root system conductivity can be achieved by increasing the lateral branching density, changing root diameters or increasing the primary root growth rates (Klein et al., 2020;

Meunier et al., 2020). Again, these different strategies are tied to different costs and benefits, creating multiple redundant development pathways that could potentially be leveraged in different environments.

Root systems also display striking **heterogeneity** in both structure and function. Different root classes **vary in their growth angles**, **anatomy**, **final length**, **lifespan** and **hydraulic conductance**. For example, steeper roots (e.g., the DEEPER ROOTING 1 trait in rice) can access groundwater reserves (Uga et al., 2013). Within a single genotype, differences in suberization, xylem architecture and aquaporin expression can be observed both along and between lateral roots (Bagniewska-Zadworna et al., 2012; De Rybel et al., 2016; Wang et al., 2020). Developmental zonation further introduces heterogeneity, as younger root zones are more permeable and responsive to environmental cues than mature zones. This spatial variation in function allows plants to exploit a range of soil microenvironments and reduces reliance on any single root portion. Even within roots of the same topological order, random heterogeneity has been observed and quantified in several plant species (Passot et al., 2018). It has been hypothesized that such diversity could be a functional adaptation for temporally and spatially variable and unpredictable soil environments (Muller et al., 2019).

Anyone who has ventured into the study of root architecture knows how incredibly **plastic** it can be. While in a research setting, this plasticity is painful to work with and requires many replications to ensure that the observed effect is indeed caused by specific treatment or genotype. In 'the wild', plasticity of root architecture allows the plant to tailor its response to soil water heterogeneity, enabling adaptive reallocation of growth and function (Bao et al., 2014; Dietrich et al., 2017; Orman-Ligeza et al., 2018; Orosa-Puente et al., 2018; Scharwies et al., 2024). In the field, drought can lead to reduced lateral root density near the soil surface and promote deeper root elongation in species like maize and chickpea (Kashiwagi et al., 2005; Zhan et al., 2015). On a smaller scale, water availability gradients guide the root tip towards available water (hydrotropism) (Dietrich et al., 2017). Even outside of the plastic root tip, water gradients can modulate emergence and elongation of lateral roots, where **hydropatterning** promotes lateral root development in the areas with high moisture content (Bao et al., 2014; Orosa-Puente et al., 2018; Scharwies et al., 2024) while **xerobranching** represses lateral root formation when growing through a waterless gap (Mehra et al., 2022; Orman-Ligeza et al., 2018; Orosa-Puente et al., 2018). Plasticity is also evident in root growth arrest and reactivation cycles in response to fluctuating water potential over time. When water potential drops, newly developed lateral roots enter a temporary quiescence phase, which protects the growing tip in the long term (Geng et al., 2013; Rowe et al., 2016). Mutants lacking this pause tend to cease their growth entirely at a later stage (Geng et al., 2013), suggesting that early slowdown supports later recovery. While prolonged stress can cause irreversible damage, the line between reversible and irreversible changes in root systems is not well defined. Unlike above-ground traits, where water stress recovery and damage are better studied (Feller, 2016; Gilgen & Feller, 2014; Liu & Bennett, 2011), **reversibility of root responses (the capacity to revert to pre-stress state) is less understood**. Future work should focus on tracking how and when root growth can bounce back, to better link plasticity with resilience.

In the past, root ideotypes such as the 'steep, cheap and deep' model proposed by Lynch (2013) have become a target for breeding programmes seeking drought-tolerant crops. However,

these ideotypes are unlikely to perform well across various drought stress scenarios (Tardieu, 2012). Determining the degree to which the plant can maintain root architecture robustness through integration of empirical root architecture studies with modelling approaches, such as Functional Structural Plant Models (OpenSimRoot (Postma et al., 2017), CRootBox (Schnepf et al., 2018) or MECHA (Couvreur et al., 2018), will provide improved insights into plant performance across various water deprivation scenarios and predict their resilience.

## 5. Beyond the reference: (Pan)genomic basis of robust water transport

The genomic architecture of plants underpins their capacity to construct and regulate efficient and robust water transport systems. Across evolutionary timescales, duplications, transposon activity and gene family expansions have contributed to the emergence of **redundancy**, **heterogeneity** and **plasticity** at the molecular level. These features are evident in the transcriptional regulation of water transport traits, the structural diversity of gene families related to hydraulics and the inter- and intra-species variation uncovered through pan-genomic studies.

**Redundancy** at the genetic level often arises through **whole-genome** and **small-scale duplications**, which have played a pivotal role in land plant evolution (Jiao & Paterson, 2014; Kondrashov, 2012; Panchy et al., 2016). Many key regulators of water transport, such as aquaporins, cell wall-modifying enzymes and suberin biosynthesis genes, are encoded by multigene families with partially overlapping functions (Cohen et al., 2013; Feng et al., 2022; Sasidharan et al., 2011). For example, the *Arabidopsis* genome contains over 30 aquaporin genes grouped into subfamilies (PIPs, TIPs, NIPs and SIPs), several of which are co-expressed and compensate for one another during abiotic stress (Maurel et al., 2015). This molecular redundancy allows for functional backup in case of mutation or environmental perturbation, increasing the robustness of hydraulic control. Gene duplication also facilitates sub-functionalization, where duplicate genes specialize in spatial or temporal expression – contributing further to robustness by distributing risk across tissue zones or environmental contexts. Uncovering functional gene redundancy in water transport requires approaches beyond single-gene knockouts, which often fail to reveal phenotypes due to compensation by paralogs. Tools like MULTIKNOCK or CRISPR-Combo, where guide RNAs are designed to target multiple close orthologs, enabling simultaneous inactivation of multiple related genes (Berman et al., 2025; Hu et al., 2023; Pan & Qi, 2023), can help us identify the minimal redundancy needed for stress resilience. Cell-type-specific expression analysis and tissue-targeted gene editing can further clarify whether gene copies operate redundantly or have distinct roles in different spatial or temporal contexts. Meanwhile, the mobility of transcripts and proteins across cell layers adds another layer of buffering – allowing gene products to act beyond their site of origin. Combining multi-gene knockouts, spatial transcriptomics and tracking of mobile RNAs and proteins will allow us to better understand which redundancies are biologically necessary and how they contribute to the robustness of plant water transport networks.

**Heterogeneity** at the genomic level is reflected in **allelic variation**, which drives differential performance under water stress conditions. For instance, twofold natural variation in root water conductivity was observed among 15 *Arabidopsis* accessions (Sutka et al., 2011), where aquaporin-dependent and independent processes showed significant and independent variation. Root suberization was also observed to vary widely among *Arabidopsis*

accessions (Feng et al., 2022). Accessions collected from drier environments tend to have higher suberin content and different suberin composition. These patterns suggest that water transport components are not genetically fixed traits, but rather their allelic variation is subject to environmental adaptation.

Genetic heterogeneity extends beyond single-nucleotide variants to the **structural architecture** of the genome itself. Pan-genomic analyses – comparing the full gene repertoire across multiple accessions or cultivars – have revealed that up to 20% of genes in species like maize, rice and soybean may be dispensable or variably present, often including genes related to environmental adaptation (Golicz et al., 2016). Here, we use **accessory** and **dispensable** genomes interchangeably to refer to the genes present in some but not all individuals of the species, in contrast to the **core genome**, which is shared across all accessions. Some research teams further distinguish accessory genes as conditionally adaptive and dispensable genes as non-essential; however, detailed functional validation of this functionality is yet to be experimentally validated through gene knockouts in multiple backgrounds. At this moment, it is assumed that both accessory and dispensable genome elements contribute to genomic heterogeneity and reflect the same underlying evolutionary flexibility of the plant genome. Interestingly, **accessory genomes** are typically enriched in genes associated with stress perception, signalling and secondary metabolism, many of which modulate water transport and root development in response to water deprivation (Gao et al., 2019; Liang et al., 2023; Petereit et al., 2022; Schreiber et al., 2024; Wang et al., 2023). These variably present genes, therefore, form an adaptive layer of genetic redundancy and functional diversity that can be mobilized under fluctuating environments.

Structural variants, such as copy number variants and the presence/absence of root development regulators, may mediate robustness in water-limited environments, especially in crops domesticated across a wide geographic range (Hirsch et al., 2014). These observations point to a wide range of water transport strategies embedded within a species. Rather than being liabilities, these genomic differences represent a reservoir of adaptive potential that can be harnessed for both resilience and productivity. Modern breeding programmes already capitalize on this principle. The development of F1 hybrids often relies on combining complementary accessory genomes from elite parental lines, allowing hybrid vigour to emerge from underlying genetic heterogeneity. Leveraging pan-genomic analyses is currently utilized in designing genetic combinations with minimal deleterious genomic variants (Cheng et al., 2025; Sun et al., 2025), but combining complementary accessory genomes can further maximize hybrid vigour based on genomic heterogeneity. However, the use of genomic heterogeneity for agricultural purposes is not new. For centuries, indigenous and traditional farming systems have relied on managing intraspecific diversity in small-scale, low-input agroecosystems (Altieri & Nicholls, 2017; Barnaud et al., 2009; Mutegi et al., 2012; Wang et al., 2016). These systems – spanning polycultures, landraces and mixed seed banks – have been shown to stabilize yields, reduce environmental risk and sustain food security through environmental unpredictability (Altieri, 1999; Andreotti et al., 2023; Beyene et al., 2006; Mulvany, 2021; Vandermeer & Perfecto, 2007). Together, both modern and traditional strategies highlight the functional value of genetic diversity in sustaining water transport under stress. Natural diversity panels, landrace collections and pan-genomic resources offer essential tools to identify and conserve this variation. In the context of increasing climate variability, breeding for resilience will depend not only on identifying the 'best' water

transport alleles but also on embracing the full spectrum of genetic solutions that plants have evolved across landscapes and cultures.

Beyond structural variation, plants exhibit **transcriptional plasticity** – the ability to reprogramme gene expression in response to changing water availability. This includes rapid upregulation of hydraulic genes like aquaporins, suberin biosynthesis genes (e.g., *GPAT5* and *CYP86A1*) and stress-responsive transcription factors (e.g., *DREB*, *NAC* and *MYB*) (Barberon et al., 2016; Dinneny, 2019). Plastic expression responses allow plants to adjust their water transport capacity at a moment's notice. Epigenetic mechanisms, including DNA methylation, histone modifications and non-coding RNAs, also contribute to gene expression plasticity. Stress-induced epigenetic memory may prime future responses and enhance robustness over time or across generations (Crisp et al., 2016). It is important to keep in mind that at the transcriptional level, heterogeneity and plasticity often co-occur, as environmentally responsive expression programmes can generate population-wide variation even under uniform conditions.

Understanding the genetic underpinnings of robustness in water transport systems requires moving towards **genomic diversity-aware models**. Pan-genomes and expression atlases constructed across various growth conditions (Kajala et al., 2021) provide critical tools for identifying robustness-associated alleles and transcriptional responses across genotypes and environments. Combining genomic data with phenotyping and environmental modelling will be key for identifying trait-stabilizing loci and deploying them in breeding programmes focused on stress resilience.

Distinguishing the core and accessory genomes provides a useful framework for understanding how robustness operates at the genetic level. The *core genome* preserves essential hydraulic functions that sustain baseline performance, while the *accessory genome* provides a flexible repertoire of stress-adaptive genes that expand a species' capacity to respond to environmental variability. This dual structure mirrors the broader principles of robustness, where redundancy safeguards function, heterogeneity diversifies responses and plasticity enables dynamic reconfiguration across scales.

While architecture and anatomy define the spatial and physical substrate for water transport, their regulation is ultimately rooted in the genome. The extent of anatomical and architectural plasticity that a plant can express under stress depends on its genetic circuitry and the regulatory loops controlling gene expression. Integrating environmental cues, hormonal signalling pathways, epigenetic modifications and allelic variation in structural and regulatory genes into these regulatory networks will help elucidate the feedback loops linking genomic, anatomical and architectural scales. In this integrated view, robustness emerges not from any single level but from their interaction: Genetic diversity shapes anatomical potential, which in turn constrains or enables architectural responses.

## 6. Measuring robustness: Quantifying the invisible buffer

While robustness is a central property of biological systems, it is not a single measurable trait but an emergent property resulting from dynamic interactions across components. In plant water transport systems, robustness reflects the capacity to maintain hydraulic function in the face of environmental, structural or genetic perturbations. Capturing this capacity requires a shift from traditional trait measurement towards **integrative, multi-scalar frameworks** that assess not only performance but also **resilience, adaptability and failure modes**. Here we propose a roadmap for quantifying the three core contributors to robustness – **redundancy**, **heterogeneity** and **plasticity** – across anatomical, physiological and phenotypic dimensions.

**Redundancy** can be measured by assessing the **degree of trait overlap across tissues, organs or genotypes** that contribute to similar functions. For instance:

- **Xylem redundancy** can be estimated using the *vessel grouping index*, quantifying the number of vessels per group versus isolated vessels in cross-sections (Lens et al., 2011). High grouping values suggest greater hydraulic backup in case of embolism.
- **Root redundancy** may be assessed by modelling water flow through Root System Architecture using functional-structural models like OpenSimRoot (Postma et al., 2017), CRootBox (Schnepf et al., 2018) or MECHA (Couvreur et al., 2018). By taking explicitly into account the architectural and anatomical complexity of complete root structures, these models can simulate water movement in the complete soil-plant domain for contrasting combinations of traits and predict alternative water uptake pathways under perturbation.
- **Genomic redundancy** is measured via gene family expansions and functional overlap – for example, through transcript co-expression analysis under varying environmental conditions.
- **Functionally**, robustness based on the redundancy can be estimated by comparing root performance decline after perturbation (e.g., vessel blockage or root pruning), providing direct evidence of buffered function.

**Heterogeneity** refers to the **diversity of trait values** within a given plant or population. This can be measured through:

- **Anatomical heterogeneity**: Use of image analysis (e.g., X-ray Computer Tomography and microscopy) to quantify variation in xylem diameter, pit membrane structure or lignin distribution within organs or across genotypes. The main bottleneck for such quantification lies in the lack of throughput for anatomical quantification, except for a few specialized methods (such as the Laser Ablation Tomography (Strock et al., 2019). To fully access and quantify anatomical heterogeneity, there is a need for more accessible methods that could be used at scale.
- **Root trait heterogeneity**: Calculating heterogeneity across root types or zones for traits like branching density, angle or suberization intensity (Rellán-Álvarez et al., 2015) is necessary to assess the patterns and allow for the identification of emergent properties. As discussed in detail by Muller et al. (2019), lateral roots of the same order can have strikingly different functions (absorptive vs. transportive). To fully embrace root trait heterogeneity means moving away from comparing trait averages to looking at their spread within genotypes. Such methods have long been used in the ecology field, such as the Shannon indices (Chegdali et al., 2024; Sun et al., 2019) or the coefficients of variation. As for the anatomical traits, there is also a need to move away from

global metrics, such a total root length or total biomass, that, despite their interest, can hide differences in individual developmental processes and completely restrict our ability to detect heterogeneous patterns.

- **Gene expression heterogeneity**: Single-cell/-nuclei Ribonucleic Acid sequencing or spatial transcriptomics across different conditions and time points can help to map divergent responses in root cells during stress. Time-course transcriptomics can be used to explore gene expression plasticity, as it captures dynamic temporal reprogramming of the transcriptome in response to stress. However, when examined across individuals or cell types, such temporal changes also contribute to observed heterogeneity – illustrating how plasticity and heterogeneity can overlap depending on the scale and context of measurement. As with all experiments, providing exhaustive metadata accompanying each RNA-seq dataset will allow for more exhaustive cross-comparisons and improved insights.

Incorporating trait correlation matrices or multivariate Principal Component Analysis can further help identify **uncoupled versus redundant components** within the studied system, where **uncorrelated** traits contribute to robustness through diversification rather than overlap. One example of such traits would be the number of nodal roots (in cereals) and the number of meta-xylem vessels. Both traits effect the capacity of the root system to uptake water, but are controlled by completely different developmental processes.

**Plasticity** is the **ability of traits to change in response to environmental stimuli**, and its quantification is key to understanding system responsiveness:

- **Reaction norms** plot trait values (e.g., root depth, vessel density and transpiration rate) across environmental gradients, and can be fitted with linear or nonlinear models to extract slope (sensitivity) and range (flexibility) (Nicotra et al., 2010).
- **Time-course phenotyping** (e.g., gravimetric platforms, thermal cameras and automated imaging) captures **response dynamics** to stress onset and recovery, providing metrics like lag time, peak shift and return-to-baseline rate (Tardieu et al., 2017). Similarly, time-course transcriptomic approaches provide complementary insight into molecular plasticity, revealing transient and reversible expression changes that underlie physiological adjustments to fluctuating environments.
- **Plasticity indices**, such as the **Relative Distance Plasticity Index** or **Plasticity Stability Index**, quantify the extent and consistency of trait changes across multiple conditions (Valladares et al., 2006).
- Finally, **better and finer characterization of environmental variables** is needed to be able to characterize the direct environment of the different plant organs and their reactions to it.

Critically, plasticity must be distinguished from mere variability – it is **predictable, responsive and often adaptive**. Both controlled-environment experiments and field heterogeneity trials provide necessary insights, although integration across them remains challenging.

Applying robustness metrics for plant water transport systems in real-world, large-scale field settings presents several practical challenges. Environmental heterogeneity, such as variable microclimates, soil conditions and biotic interactions, introduces noise and unpredictability, making it difficult to generalize metrics developed under controlled conditions. Data collection remains labour-intensive, destructive or spatially/temporally limited, complicating efforts to capture whole-plant or ecosystem-level responses (and this is especially true when dealing with belowground data). Logistical constraints, including frequent recalibration, human oversight and integration with existing infrastructure, further hinder operational feasibility. Additionally, cost and expertise gaps, such as the need for advanced technical training or financial resources, can disproportionately affect smaller organizations or developing regions. These challenges highlight the need for adaptive, simplified metrics, automated tools and collaborative frameworks to bridge the gap between theoretical robustness and practical, scalable application in agriculture.

## 7. Conclusions and future perspectives

As plants contend with increasingly erratic climates, the ability to maintain water transport under stress is critical – not just for survival, but for sustained productivity. Robustness is not a fixed trait but an **emergent systems property**, shaped by both evolutionary legacy and developmental responsiveness. Importantly, robustness does not necessarily oppose efficiency – many of the most resilient systems are efficient *because* they are fault-tolerant, adaptable and context-sensitive.

Within this framework, **heterogeneity** and **plasticity** operate along a continuum rather than as discrete phenomena. Heterogeneity provides the structural and genetic variation that buffers plants against perturbation, while plasticity enables that variation to be dynamically reconfigured in response to environmental change. Recognizing their overlap highlights that resilience emerges not simply from diversity itself, but from the capacity of biological systems to modulate and reorganize that diversity when challenged.

Understanding how plants negotiate this trade-off requires dissecting water transport not just as a physiological process but as a **networked system** embedded in time, space and environment. Looking forward, we envision a framework where plant hydraulic robustness is not just described but **quantitatively predicted** – integrating phenotypic plasticity, trait covariance and genetic diversity into models of plant performance. Such a framework could combine mechanistic and data-driven approaches, where functional–structural plant models (e.g., OpenSimRoot, MECHA or CRootBox (Postma et al. 2017; Schnepf et al. 2018; Couvreur et al. 2018) are parameterized with experimentally derived traits from high-resolution phenotyping and time-course transcriptomics. Linking these models to gene regulatory and hormonal networks would enable the prediction of how genomic variation shapes architectural and anatomical adjustments under fluctuating conditions. Incorporating trait covariance and environmental data from population studies would further allow the identification of trait combinations that stabilize water transport across stress scenarios, guiding the design of genotypes with predictable resilience. Ultimately, this integration of modelling, empirical data and biological insight would connect molecular diversity to emergent hydraulic behaviour – transforming our understanding of robustness from a conceptual framework into a predictive biological tool. The fusion of high-resolution imaging, multi-omics, AI-powered and knowledge-based modelling holds great promise for unlocking the principles of resilience embedded in plant water transport networks.

As agriculture enters a period of urgent adaptation, recognizing that robustness is as valuable as raw productivity will be essential. Through interdisciplinary approaches, we can move from *observing* robustness to *designing* for it – guiding the development of crops that thrive not just in the average field, but in the face of tomorrow's uncertainty.

**Open peer review.** To view the open peer review materials for this article, please visit http://doi.org/10.1017/qpb.2025.10035.

**Competing interest.** The authors declare none.

**Data availability statement.** This review manuscript does not produce any data or code that would need to be made available.

**Author contributions.** MMJ, GL and AC discussed the concepts constituting the body of the review. MMJ drafted the initial version of the review, which was subsequently reviewed by GL and AC. All authors have contributed to the revision of the manuscript.

**Funding statement.** M.M.J. and A.C. are co-funded by the National Science Foundation (NSF IOS and Mathematical Biology #2244735). G.L. is co-funded by the European Union (ERC grant 101125638).

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
