## [Reviewer Report]

This is a very intersting manuscript on an important topic.

Just a few minor points:

In the section ‘Building belowground robustness through root system architecture’ section there are a few points where terminology can be clarified. For example in paragraph 1 and 2 of that section root types are listed e.g. ‘primary, lateral, adventitious, nodal, and crown’. E.g. crown roots, brace roots, and multiple nodal root whorls. Nodal and crown are types of adventitious roots. It would be useful to refine this list to not be redundant. In addition, ‘multiple nodal root whorls’ could be changed to ‘crown and brace roots emerging from several stem nodes’ or something similar.

Page 7 ‘within a single genotype, lateral roots and their segments’ Unclear what ‘and their segments’ means. In the same paragraph, what is ‘root sector’

Page 7 (same paragraph as comment above) is ‘random heterogeneity’ really random in this case?

Page 8, the term ‘reversibility’ should be clarified. Reversible plasticity has been used to define several different phenomenons, e.g. changes in the root phenotype at the growing root tip when stress is released or changes in the same tissue (reverting back to the original phenotype in the same tissue)

While the review proposes several metrics for measuring robustness, it might be helpful to include a more in-depth discussion on the practical challenges and limitations of applying these methods at scale, especially in a field setting. Expanding on these practical difficulties would provide a more realistic roadmap for future research.

The review discusses how different root developmental processes can achieve similar functions (redundancy) and how some root traits, such as root depth, are plastic in response to drought. It could further elaborate on the interplay between these three components. For example, how does genetic influence the degree of anatomical plasticity in response to stress?

The authors could also consider modifying the figure in the manuscript to show that architecture, anatomy, and genetics also interact!

---

## [Reviewer Report]

In this review article, Julkowska et al. discuss the concept of robustness in plant water transport, arguing that robustness—defined as the ability to maintain function under adverse conditions—rather than efficiency stricto sensu may be a more relevant target to sustain agricultural production in the future. The concept of robustness is broken down into three subcomponents: redundancy, heterogeneity, and plasticity. The authors describe how each of these components may be expressed at anatomical, architectural, and genomic scales. Overall, this is an original and nice review that is enjoyable to read.

The authors may wish to consider the following suggestions to further improve the manuscript:

-The distinction between heterogeneity and plasticity is not always entirely clear. This is evident on Page 5 under “Xylem heterogeneity” and on Page 14 under “Gene expression heterogeneity.” It might be more straightforward to explicitly acknowledge that these concepts can overlap in certain contexts.

-Page 5: The method for calculating the vessel grouping index should be explained more clearly.

-Page 10: The distinction between the dispensable and accessory genome should be clarified. The terminology could be made more consistent to avoid confusion.

-Page 14: Should time-course transcriptomics also be considered under plasticity? This is briefly mentioned in the section on gene expression heterogeneity and highlights the difficulties to distinguish between the two processes at times.

-Page 14: The statement “where uncorrelated traits contribute to robustness through diversification rather than overlap” is interesting. The authors could expand on this idea to explain more concretely how uncorrelated traits could be interpreted.

-Page 15: The statement “we envision a framework where plant hydraulic robustness is quantitatively predicted—integrating phenotypic plasticity, trait covariance, and genetic diversity into models of plant performance” is interesting but vague. Space might be limited, but providing additional insight into how such a framework could be constructed would be valuable for readers.

---

## [Editor Report]

Dear Prof. Julkowska,

Both reviewers have welcomed the review and recognised its importance to the community. They have raised a few minor comments, which I believe will improve the manuscript and make it ready for publication. We would be happy to consider a revised version that addresses these points.

All the best, 

Priya Ramakrishna

---

## [Reviewer Report]

I thank the authors for considering my previous comments. In particular, the authors have clarified their views on how heterogeneity and plasticity can overlap and have clarified some previously unclear statements. As a minor comment, genomic heterogeneity does not translate into a straightforward visualisation in the Figure 1. Appart from that, I consider that the paper has improved and is now in an acceptable form for publication.

---

## [Editor Report]

Dear Dr. Julkowska,

Thank you for the careful revision of your manuscript. All pending issues have been resolved. Thank you and the co-authors again for your valuable contribution.

Best regards,

Priya